# Fully Printed Zinc Oxide Electrolyte-Gated Transistors on Paper

**DOI:** 10.3390/nano9020169

**Published:** 2019-01-30

**Authors:** José Tiago Carvalho, Viorel Dubceac, Paul Grey, Inês Cunha, Elvira Fortunato, Rodrigo Martins, Andre Clausner, Ehrenfried Zschech, Luís Pereira

**Affiliations:** 1CENIMAT/I3N, Departamento de Ciência dos Materiais, Faculdade de Ciências e Tecnologia, FCT, Universidade Nova de Lisboa and CEMOP-UNINOVA, Campus da Caparica, 2829-516 Caparica, Portugal; jt.carvalho@campus.fct.unl.pt (J.T.C.); v.dubceac@campus.fct.unl.pt (V.D.); paul16_grey@yahoo.de (P.G.); i.cunha@campus.fct.unl.pt (I.C.); emf@fct.unl.pt (E.F.); rfpm@fct.unl.pt (R.M.); 2Fraunhofer Institute for Ceramic Technologies and Systems (IKTS), 01109 Dresden, Germany; andre.clausner@ikts.fraunhofer.de (A.C.); ehrenfried.zschech@ikts.fraunhofer.de (E.Z.)

**Keywords:** zinc oxide, nanoparticles, paper transistors, printed electronics, electrolyte-gated transistors

## Abstract

Fully printed and flexible inorganic electrolyte gated transistors (EGTs) on paper with a channel layer based on an interconnected zinc oxide (ZnO) nanoparticle matrix are reported in this work. The required rheological properties and good layer formation after printing are obtained using an eco-friendly binder such as ethyl cellulose (EC) to disperse the ZnO nanoparticles. Fully printed devices on glass substrates using a composite solid polymer electrolyte as gate dielectric exhibit saturation mobility above 5 cm^2^ V^−1^ s^−1^ after annealing at 350 °C. Proper optimization of the nanoparticle content in the ink allows for the formation of a ZnO channel layer at a maximum annealing temperature of 150 °C, compatible with paper substrates. These devices show low operation voltages, with a subthreshold slope of 0.21 V dec^−1^, a turn on voltage of 1.90 V, a saturation mobility of 0.07 cm^2^ V^−1^ s^−1^ and an I_on_/I_off_ ratio of more than three orders of magnitude.

## 1. Introduction

Printed electronics (PE) has attracted strong interest among researchers aiming to introduce simpler fabrication techniques for low cost consumer electronics. PE uses well established processes, such as flexo, screen, rotary-screen, inkjet, and off-set printing, and the use of lithography and subtractive etching steps is avoided [1]. These processes offer unique large area processing capability and high throughput on flexible substrates by roll-to-roll (R2R) processes, and it opens a path toward ultra-cost-effective, flexible, and environmentally friendly electronic devices, demanded by modern society standards.

Electrolyte-gated transistors (EGTs) have attracted great interest for flexible electronics and biosensing applications over the last decade [2]. In EGTs, an electrolyte is used as a gate dielectric between the gate electrode and the channel layer. Thus, applying a voltage to the gate electrode causes ion migration in the electrolyte, leading to the formation of an electric double layer (EDL) at the electrolyte/channel interface. This results in large gate capacitance (virtually independent of the thickness) in the order of 1–10 μF cm^−2^, enabling charge accumulation into the semiconductor at low gate voltages [3]. This makes EGTs very attractive for PE, where thin insulating dielectric layers capable of providing high gate capacitance are difficult to obtain by printing techniques at temperatures compatible with paper substrates [4,5,6,7]. Moreover, EGTs are highly sensitive to the ionic species that may contact or exist in the electrolyte. Due to their low operation voltages (<2 V) EGTs are highly desirable for biosensing applications, i.e., to determine and to quantify biological molecules inside aqueous media [8,9,10].

Solution-processed inorganic semiconductors used in printed EGTs have advantages over organic ones, such as higher intrinsic carrier mobility and stability [11,12]. However, in comparison to organic alternatives, high temperatures are needed to convert the precursors and to synthesize inorganic semiconductors from solution as well as to remove binders or stabilizers from the printed semiconductor ink [13]. These so-called burn-out temperatures could compromise their application on paper substrates. In this context, oxide semiconductor nanoparticle (NP)-based layers are an alternative, avoiding the need of high temperature and long or complex annealing processes to convert a precursor and to form a semiconducting layer. 

This study reports the development of fully printed EGTs, based on an interconnected ZnO nanoparticle matrix, arranged in a staggered-top gate architecture. Ethyl cellulose (EC), a cellulose derivative, has been proven to be an indispensable binder in the developed inks, providing the necessary rheological properties for screen printing. Additionally, this eco-friendly biopolymer provides excellent printability and leveling [14] combined with excellent adhesion to paper substrates. Proper formulation of the ZnO printable ink allows one to process these devices at temperatures below 150 °C, totally compatible with paper substrates. 

## 2. Materials and Methods

ZnO ink formulation for screen-printing: The semiconductor ink consists of a dispersion of ZnO NPs in a cellulose derivative, the ethyl cellulose (EC). ZnO nanopowder (Sigma-Aldrich, St. Louis, MO, USA, <100 nm particle size) was dispersed with different concentrations (10 wt % ZnO10 and 40 wt % ZnO40) into a solution of 5 wt % EC (Sigma-Aldrich, St. Louis, MO, USA), viscosity 300 cP, extent of labeling: 48% ethoxyl) in a solvent mixture of toluene/ethanol (80:20 *v*/*v*) supplied from Merck (Darmstadt, Germany), ≥99% and Fisher Scientific (Pittsburgh, PA, USA), respectively. The resultant mixture was stirred for 12 h to obtain a well dispersed and homogeneous white viscous solution. Finally, the prepared inks were stored in a refrigerator at 3 °C, due to the high volatility of the components, until use.

Composite solid polymer electrolyte (CSPE) ink formulation for screen-printing: The lithium-based polymer electrolyte ink is composed of a mixture of succinonitrile (Aldrich, St. Louis, MO, USA), an acrylic thermoplastic resin (TB 3003 K, ThreeBond, Saint-Ouen-l’Aumône, France), a lithium salt (LiClO_4_, Sigma-Aldrich), and titanium dioxide NPs (TiO_2_, Rockwood Pigments NA, Inc., Beltsville, MD, USA). All proceeding steps were carried out at 60 °C and relative humidity of 50%.

Electrolyte-gated transistor fabrication and characterization: The fully printed ZnO NP EGTs were produced with a staggered-top gate structure onto glass (Marienfeld) and paper substrates, by a custom-made screen-printing. All materials were used without any further purification. Carbon interdigital source/drain (S/D) electrodes (effective W/L = 39725 µm/330 µm ≈ 120) were printed by a polyester mesh model 120-34, using a carbon paste (TU-10s, Asahi Chemical Research Laboratory Co., Ltd., Utsuki-cho, Hachioji, Tokyo) dried at 150 °C for 30 min in air. An average thickness of 5.46 ± 0.51 µm, through a single printing step, was achieved. For the channel layer, ZnO10 and ZnO40 inks were printed through a mesh model 77-55 with a single printing step, achieving an average thickness of 1.39 ± 0.14 µm and 4.60 ± 0.72 µm, respectively. The same mesh was used to print the CSPE layer (average thickness of 18.43 ± 1.23 µm), which was cured at 70 °C for 5 min, followed by UV-exposure for 5 min. Lastly, the silver gate electrode was printed via a 120-34 mesh (average thickness of 1.78 ± 0.60 µm). For that, silver ink (PE-AG-530, from Conductive Compounds, Inc., Hudson, NH, USA) was printed and dried at 70 °C for 5 min, in air.

Prior to printing steps, glass substrates were cleaned in an ultrasonic bath for 5 min, first in acetone and then in isopropanol, after that rinsed off in deionized water (Millipore) and then dried using nitrogen. Regarding the paper substrates, regular printing paper and nanocellulose paper was based on microfibrillated cellulose (referred to as MFC Kraft). MFC Kraft was produced from a Kraft pulp subjected to a mechanical treatment followed by six homogenizing steps at 1500 bar. The nanocellulose paper was prepared by slow casting and evaporation of water from 20 mL of MFC suspension (2 wt % MFC) under ambient conditions in polystyrene Petri dishes (85 mm diameter). The resulting paper substrates had a thickness of approximately 62.4 µm, estimated from the average of five measurements made using a Mitutoyo digital micrometer. 

The thickness of the screen-printed layers was measured with an Ambios XP-Plus 200 Stylus profilometer. The channel length (L) and width (W) of the EGTs were measured using a Leica IC80 HD microscope and LAS V4.3 software. The EGTs were electrically analyzed in the dark at room temperature using a microprobe station (M150 Cascade Microtech, Beaverton, OR, USA) connected to a semiconductor parameter analyzer (4155C Agilent, Santa Clara, CA, USA) controlled by the software Metrics ICS.

Nanoindentation experiments were performed using a Hysitron TI 950 Triboindenter tool with a Berkovich indenter on three different sample regions, with the goal to characterize the films mechanically. For each region, the indents were performed with a maximum load, P_max_, of 1000 µN, 2500 µN, and 5000 µN. For each P_max_, 20 indents were made, with a 5 s loading time, a 10 s creep time, and finally a 5 s unloading time.

## 3. Results and Discussion

Figure 1a depicts the process steps of the fully printed EGTs arranged in a staggered-top gate architecture. Figure 1b,d show the topographical SEM surface image of screen-printed layers on glass starting from ZnO inks with different NPs content (for details regarding ink formulation, please see “Materials and Methods” section). The morphology is similar, despite the visible agglomerates for the lowest ZnO content (Figure 1c,e). The RMS surface roughness values obtained by AFM (Appendix A) are 115.1 nm and 132.8 nm for layers printed from ZnO10 (10 wt % ZnO NPs) and ZnO40 (40 wt % ZnO NPs) inks, respectively. It is known that high surface roughness can heavily influence the interface quality, compromising the device performance [15]. Nonetheless, the rheologic characteristics of the ink developed here provided a good packing and self-leveling of the NPs, resulting in excellent film compactness and substrate coverage, ensuring continuous conduction paths, as required for fully functional devices.

Since the target is to fabricate EGTs on paper substrates, the mechanical properties of individual ZnO40 screen-printed films were studied after being submitted to bending cycles. Appendix A shows that the electrical resistance of the screen-printed ZnO40 films on paper continuously increases after 100, 500, and 1000 bending cycles. Crack formation (see Appendix A) observed after 1000 bending cycles helps to explain this. Nanoindentation was performed in three different regions of the ZnO screen-printed layers subjected to the 1000 bending cycles to determine Young´s modulus (E) and the indentation hardness (H) (see Appendix A). For the analysis, the maximum penetration depth (h_c_), was set around 2 µm, i.e., less than 10% of the thickness of the layer, to avoid effects of the paper substrate [16]. Both E and H decrease with increased h_c_, which is related with the fluence of the non-sintered ZnO NPs during the indentation. The non-ideal cohesion between them also explains why the E and H are consistently lower after bending cycles.

The ionic and electrochemical response of the CSPE used in this work was determined using electrochemical impedance spectroscopy (EIS). For further details about CSPE characterization please see Appendix A. Figure 2a,b show the experimental data and the fitting obtained from the equivalent circuit model (ECM) for both Bode and Nyquist plots. EDL formation (stabilizing effective capacitance, C_eff_) occurs for frequencies below 25 Hz, determining the transition from the resistive to the capacitive regime (phase angle equal to −45°). This is normally also a good indicator about the cut-off frequency [17], which is usually a limitation of the EGTs, namely those using solid-state electrolytes.

C_eff_ gradually stabilizes at a value of about 20 µF cm^−2^ for low frequencies. Some authors use this value as the C_DL_ for further calculations, which is a conservative approach if it is not possible to fit the impedance data with the model, leading to underestimated mobility values, for instance. However, it does not consider contact resistances or the surface inhomogeneities, leading to erroneous values. Here we use Equation (1), proposed by Jovic et al. [18].
(1)$$C_{\mathrm{DL}}={\left[Y_{0}R_{\mathrm{ext}}^{-\left(\alpha -1\right)}\right]}^{1/\alpha }$$
where $Y_{0}$ is linked to the capacitance of the CPE, and α is a constant (between 0 and 1), often referred to as the fractal surface character of the interface, denoting how non-ideally the CPE behaves. The value of C_DL_ was determined at 2.47 µF cm^−2^. It is important to note that countless factors are involved, such as the cell setup and dimensions, or even the ECM fitting parameters, which also have an associated error. Nevertheless, the determined values are in the typical range for the use of solid electrolytes in EGTs, i.e., 1–10 μF cm^−2^, as described by Kim et al. [2].

The transfer characteristics (drain-source current (I_DS_) vs. gate-source voltage (V_GS_)) of the fully printed ZnO NP EGTs on glass and on paper substrates are shown in Figure 3a,c, respectively. Apart from the ZnO concentration in the ink (10 and 40 wt %) and the substrate used, the device characterization was focused on the influence of the post-printing annealing temperature of the semiconducting inks. The consequent removal of the binder (EC) would ultimately lead to the desired electrical properties, given by the ZnO NPs. As evidenced by TG-DSC (see Appendix A), temperatures above 350 °C are high enough to fully degrade the binder. Taking this temperature into account, Figure 3a depicts the I_DS_–V_GS_ transfer characteristic for the ZnO10_350 °C Glass_ EGT. These devices printed on glass substrates show n-type and normally off behavior. Once in the on-state, they reach an I_DS_ value in the range of 10 µA, with an on/off current (I_on_/I_off_) ratio of three orders of magnitude, owing to the intrinsic high double layer capacitance of the CSPE [19]. On the other hand, no modulation was observed if the annealing temperature decreases below 350 °C. 

Aiming to reduce the annealing temperature, the ZnO nanoparticle concentration in the ink was increased up to 40 wt %. As depicted on Figure 3a, proper optimization of the NP content allows for electrical modulation at annealing temperatures of only 150 °C (ZnO40_150 °C Glass_). Higher NP content in the ink promotes more conductive paths between the S/D electrodes, without compromising the printability and layer formation. It is worth mentioning here the advantage of using screen-printing since it enables one to print in a wide range of viscosities, allowing the increase in the ZnO NPs content in the ink. Doing so, the resulting printed ZnO layer have lower impedance |Z|, as shown in Figure 3b, suitable for use as a channel in EGTs. Thus, by increasing the ZnO NP concentration, the impedance measured for printed patterns using the ZnO40 ink without annealing, is similar to the one measured when using ZnO10 ink after annealing at 350 °C. Consequently, the performance of EGTs based on ZnO40 ink annealed at 150 °C is similar to that obtained for devices using the ZnO10 ink annealed at 350 °C, with an on/off ratio in the same order of magnitude, a saturation mobility (µ_Sat_) of 0.02 cm^2^ (Vs)^−1^, but a higher anti-clockwise hysteresis. The saturation mobility (µ_Sat_) was determined using Equation (S1), as described in the Appendix A.

It can be inferred that, without eliminating totally the EC binder, it may behave as an extension of the electrolyte into the semiconducting layer. Previous studies have shown that cellulose can have electrolyte-like behavior in paper transistors [20,21,22,23,24]. Thus, if the EC is not totally eliminated and remains in the printed layer, electrolytic behavior is extended to the ZnO NP channel. The ZnO matrix stays intact as the current modulation is observed. However, ions can penetrate or be trapped in the EC matrix and remain in the vicinity of the semiconductor NPs, even when sweeping back to negative V_GS_, resulting in ion trapping. The hypothesis of electrolyte extension is sufficiently supported by the results obtained for the EGT based on ZnO40 ink annealed at 350 °C printed on glass (ZnO40_350°C Glass_ EGTs). Its I_DS_–V_GS_ characteristics highlight a considerable reduction in the anti-clockwise hysteresis. This fact is expected since the EC is burned-out, as already explained, so the ion trapping phenomena within the channel layer might be highly reduced. Moreover, a clear increase in both I_off_ and I_on_ of the device is visible from the same I_DS_–V_GS_ characteristics. This increase is related to an enhanced channel conductivity, resulting from the improved charge transport between the ZnO NPs due to the thermal degradation of EC. Again, the higher NP concentration in the ZnO40 ink in comparison with the ZnO10 one leads to a higher NP density in the printed films. Moreover, partial sintering of the NPs, during the burn-out process, should not be excluded, also in relation to the increasing I_Off_ and I_On_. In fact, it is known that the sintering temperatures of nanosized structures owing to the high surface-to-volume ratio are considerably lower compared to the bulk material [25]. As a result, the ZnO40_350 °C Glass_ EGT stands out with a µ_Sat_ of 5.73 cm^2^ (Vs)^−1^, an I_On_/I_Off_ ratio of almost four orders of magnitude, and reduced hysteresis, less than 1 V. The performance of these devices compares well with reported values for nanoparticle-based EGTs. Bong et al. [26] reported on ZnO transistors annealed at 280 °C, using an ion-gel as electrolyte (drop-casted) with a µ_Sat_ of 12.1 cm^2^ (Vs)^−1^ and an I_on_/I_off_ ratio of about 10^5^. Santos et al. [11] have reported spin-coated gallium-indium-zinc-oxide (GIZO) NP EGTs, with I_on_/I_off_ ratios between 10^3^ and 10^6^ and a µ_Sat_ between 6 × 10^−3^ and 1 cm^2^ (Vs)^−1^. Likewise, Dasgupta et al. [15] have reported inkjet printed indium-tin-oxide (ITO) EGTs, with an I_on_/I_off_ ratio of about 10^4^ and a mobility of 5 cm^2^ (Vs)^−1^, after annealing at 400 °C.

It is worth mentioning that the µ_Sat_ here obtained for the fully printed ZnO EGTs may be influenced by surface roughness. Consequently, it generates charge traps at the ZnO/electrolyte as well as increases the contact resistance at S/D electrodes, as suggested by Dasgupta et al. [15]. 

Finally, we could reproduce our ZnO40 EGTs on conventional printing paper (ZnO40_150 °C PP_ EGT, Figure 3c,e) and MFC Kraft substrates (ZnO40_150 °C MK_, Figure 3c,f), since they can withstand temperatures up to 150 °C. The difference in the µ_Sat_, (0.01 and 0.07 cm^2^ (Vs)^−1^) for both paper substrates, depicted in Table 1, can be explained by the influence of the paper substrates roughness. Via 3D profilometry (see Appendix A), the scanned area (500 µm × 500 µm) of the conventional printing paper and MFC Kraft yields an average surface roughness of 2.98 and 1.70 µm, respectively. As verified earlier, the high surface roughness of the substrate owing to the poor dispersion of the ZnO NPs contributes to a large interface roughness between the EGT channel and the CSPE. This result confirms conclusions drawn by Dasgupta et al. [15]. Consequently, the ineffective gating leads to non-uniform electric fields in the semiconducting layer, originating lower µ_Sat_. Moreover, increased S/D contact resistance can also affect the measured transfer characteristics and consequently the calculated carrier mobility.

All the ZnO NP EGTs reported here exhibit extremely low subthreshold slope (SS), showing the rapidness of the ion accumulation process. Additionally, the characteristic low-voltage operation between −2 and 4 V, for this device technology (EGTs), was confirmed. Besides that, the dynamic characterization of the ZnO40_150 °C MK_ EGT (see Appendix A) showed that the device operates between 0.01 and 10 Hz without reaching the cut-off, in accordance with the capacitance/frequency dependence, keeping still above one order of magnitude in the I_on_/I_off_ ratio.

## 4. Conclusions

In summary, this study reports a fully printed and flexible inorganic EGT on paper. Besides the low operation voltages, the devices have n-type behavior with an SS of 0.21 V dec^−1^, a µ_Sat_ of 0.07 cm^2^ V^−1^ s^−1^, and an I_on_/I_off_ ratio of 3.72 × 10^3^. Investigation in this field could lead to the effective roll-to-roll fabrication of printed logic circuits, with great implications for industrial printing standards, including sustainable and eco-friendly ink formulation. In this context, electrolyte gating, using printable electrolytes, is crucial for the development of printed transistors. In the near future, we foresee printed EGTs to be omnipresent in a wide range of applications, reaching from biosensors, smart packaging to wearable electronics all based on an ultra-low-cost and disposable/recyclable platforms. 

## Figures and Tables

**Figure 1 nanomaterials-09-00169-f001:**
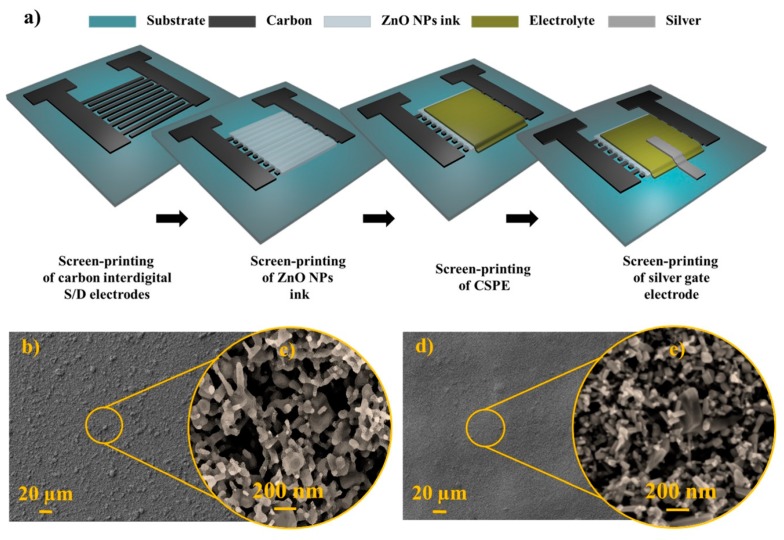
(**a**) Schematic representation of the fabrication steps for the fully printed ZnO electrolyte-gated transistors (EGTs). (**b**,**c**) Topographical view of the screen-printed layer using ZnO10 ink. (**d**,**e**) Topographical view of the screen-printed layer using ZnO40 ink.

**Figure 2 nanomaterials-09-00169-f002:**
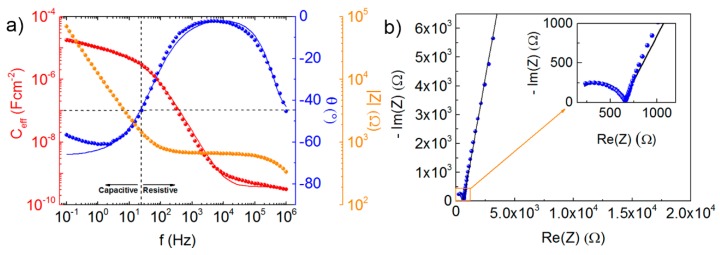
(**a**) The measured effective capacitance, phase, and impedance variation with frequency and (**b**) the respective Nyquist plot of the composite solid polymer electrolyte (CSPE).

**Figure 3 nanomaterials-09-00169-f003:**
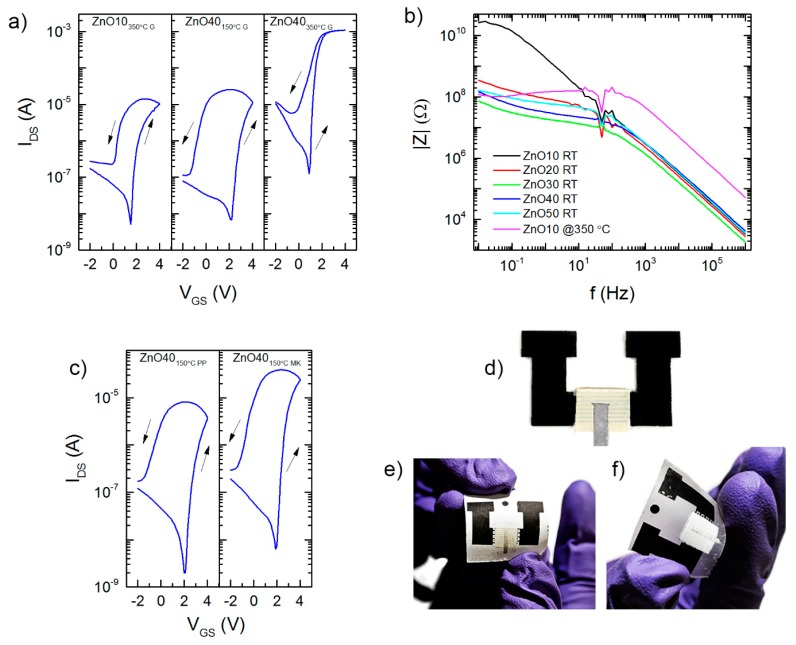
(**a**) Representative transfer characteristics (I_DS_–V_GS_) of the ZnO NP EGTs on glass, namely ZnO10 EGT annealed at 350 °C (ZnO10_350 °C Glass (G)_); the ZnO40 EGT under processing temperatures of 150 °C (ZnO40_150 °C G_) and the ZnO40 EGT annealed at 350 °C (ZnO40_350 °C G_), respectively. (**b**) Impedance |Z| of the developed ZnO NPs inks, i.e., ZnO10, -20, -30, -40, and -50 measured without annealing (RT), and for the ZnO10 ink after annealing at 350 °C. (**c**) The I_DS_–V_GS_ of the ZnO40 EGT on printing paper (ZnO40_150 °C PP_) and the ZnO40 EGT on MFC Kraft (ZnO40_150 °C MK_), both at processing temperatures of 150 °C. An image of the printed EGTs on (**d**) glass, (**e**) printing paper, and (**f**) MFC Kraft.

**Table 1 nanomaterials-09-00169-t001:** Summary of the electrical characterization of the produced fully printed ZnO NP EGTs: channel width and length ratio (W/L), the drain voltage (V_DS_), the turn on voltage (V_on_), the on-off current ratio (I_on_/I_off_), the subthreshold swing (SS), the transconductance (gm), and the saturation mobility (µ_Sat_).

Device Designation	W/L	V_DS_ (V)	V_on_ (V)	I_on_/I_off_	SS (V dec^−1)^	g_m_ (S)	µ_Sat_ (cm^2^ (Vs)^−1^)
ZnO10_350 °C G_ EGT	120	1.2	1.50	2.00 × 10^3^	0.08	2.58 × 10^−6^	0.02
ZnO40_150 °C G_ EGT	120	1.1	2.20	1.55 × 10^3^	0.11	3.55 × 10^−6^	0.02
ZnO40_350 °C G_ EGT	120	0.9	0.90	8.74 × 10^3^	0.06	8.44 × 10^−4^	5.73
ZnO40_150 °C PP_ EGT	120	1.0	2.10	1.73 × 10^3^	0.01	1.41 × 10^−6^	0.01
ZnO40_150 °C MK_ EGT	120	1.3	1.90	3.72 × 10^3^	0.21	1.08 × 10^−5^	0.07

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
