# Peer review of "Fully Printed Zinc Oxide Electrolyte-Gated Transistors on Paper"

_nanomaterials, 2019, doi:10.3390/nano9020169_

Round 1
Reviewer 1 Report
Manuscript Title: Fully-printed zinc oxide electrolyte-gated transistors on paper
The manuscript reports on the performance optimization of fully printed and flexible inorganic electrolyte gated transistors (EGT) based on interconnected ZnO nanoparticle matrix printed on flexible substrates. Although the electrochemical gating with polymer electrolyte is not a new concept the proposed technological solution allowing for scalable low-temperature processing of EGT on paper substrates has large commercialization potential. The work is original and the reported results make a valuable contribution to the field. The content of the paper matches the scope of the MDPI Nanomaterial Journal. Nevertheless, there are a several issues, which need to be addressed before the publication of the manuscript. Thus, I would recommend the publication after minor revision regarding the following comments:
Page 2, rows 63 to 70 – The description of ZnO ink formulation is not sufficiently clear. Better definition of the receipt is recommended. In example – is there a difference between the solution of 5wt% of EC (in ethanol and toluene) and the solvent mixture of ethanol and toluene?
Page 2, row 75 - How was the humidity controlled and why was it controlled?
Page 3, row 115 – Although the AFM 5x5 µm show higher surface roughness for ZnO40, the topographical views in Figs 1 (b) and (d) suggest smoother surface for ZnO10. Do the authors have an explanation for this controversy?
Page 3, row 118 – The authors claim that an effect of “self-arrangement” is an inherent property of the ink they developed. What is their definition of such “self-arrangement” and how do they explain it?
Page 3, row 124 – As the authors performed up to 1000 bending cycles (Fig S2) it is recommended that they provide an error statistic to the graph.
Page 3, row 129 – 133 The authors do not provide a proper quantitative analysis of the nanoindentation data other than to mention that “the high surface roughness and the movement of the nanoparticles due to the indentation may influence the E and H values”. They should either provide an explanation of the nanoindentation data or remove it from the manuscript if they consider it non-essential to the text.
Page 5, row 162 – Please, define all abbreviations (e.g. IDS, VGS) before using them in the text.
Page 5, row 162 – What is the difference between Fig. S7 and Fig. S8? They have identical captions!
Page 5, Table 1 - As the saturation mobility (μSat) is used as an important figure of merit the authors should explain how it is calculated and compare their figures with the literature data on flexible inorganic electrolyte gated transistors published by other authors.

Reviewer 2 Report
In the paper ‘Fully-printed zinc oxide electrolyte-gated transistors on paper’, Carvalho J. T. et al. reported a novel TFT based on ZnO nanoparticles embedded in a cellulose derivative as semiconductor.
The device has been fabricated by screen printing on glass substrates or on common paper supports and the characterization has been performed via electrolyte gating. The paper is interesting and the fabrication approach appealing but I would like the authors to address some points before accepting the manuscript.
1-The EIS analysis of the composite solid polymer electrolyte is based on the circuit reported in the supporting information which include a CPE in series with the electrochemical ‘geometric’ contribution of the material but for the reader the two parts are hard to correlate. The CPE takes into account of the two double layers at the interfaces but I don’t understand why a second resistance associate with the CDL is not present in the circuit.
It is true that the ref18 uses a CPE but they employed different circuits, can you better justify your choice?
2-Could you please explicit the values obtained from the EIS fitting?
3-It would be useful understand the behavior of the CSPE at different voltages corresponding to the operational window of the EG-TFT
4-How the measurements of fig3b are performed? The figure is not well explained in the text and the legend is misleading, RT stands for 150 degrees?
5-Did you perform any statistical analysis on the TFT response? How did you calculate the mobility?
6-On page 5, line187, the authors assume a partial intercalation of the EC binder into the semiconducting layer. Do you have any surface analysis of the semiconducting layer after the removal of the CSPE to support this assumption?
7- In the experimental section is written ‘The EGTs were electrically analyzed in the dark at room temperature’, what’s the light sensitive component?
Reviewer 3 Report
This paper reports not only fully-printed and flexible inorganic electrolyte gated transistors on paper but also eco-friendly method to disperse ZnO nanoparticles for channel layer.
They investigated electrical performance and morphology of the TFT depending on concentrations of ZnO. Also, They checked electrical resistance of the printed film on paper through bending test. Additionally, the effect of different post annealing temperature was checked
There are several points for authors to address before this manuscript is considered for publication.
1. Authors should improve resolutions of Figure S1, Figure S4, Figure S6, and Figure S9
2. This manuscript didn’t show how many devices were considered when the thickness is averaged.
3. Is there output characteristics (IDS-VDS) for the TFTs fabricated in this study?
4. As you mentioned, Ethyl Cellulose (EC) could be removed above 350 ℃. So, can you show me the TFT data fabricated above 350 ℃.
Round 2
Reviewer 2 Report
The paper is acceptable in the present form